# The Early Prediction of Kimchi Cabbage Heights Using Drone Imagery and the Long Short-Term Memory (LSTM) Model

Seung-hwan Go and Jong-hwa Park *

Department of Agricultural and Rural Engineering, Chungbuk National University, 1 Chungdae-ro, Seowon-gu, Cheongju 28644, Chungbuk, Republic of Korea; shgo916@chungbuk.ac.kr
* Correspondence: jhpak7@cbnu.ac.kr; Tel.: +82-43-261-2577

**Abstract:** Accurate and timely crop growth prediction is crucial for efficient farm management and food security, particularly given challenges like labor shortages and climate change. This study presents a novel method for the early prediction of Kimchi cabbage heights using drone imagery and a long short-term memory (LSTM) model. High-resolution drone images were used to generate a canopy height model (CHM) for estimating plant heights at various growth stages. Missing height data were interpolated using a logistic growth curve, and an LSTM model was trained on this time series data to predict the final height at harvest well before the actual harvest date. The model trained on data from 44 days after planting (DAPs) demonstrated the highest accuracy ($R^2$ = 0.83, MAE = 2.48 cm, and RMSE = 3.26 cm). Color-coded maps visualizing the predicted Kimchi cabbage heights revealed distinct growth patterns between different soil types, highlighting the model's potential for site-specific management. Considering the trade-off between accuracy and prediction timing, the model trained on DAP 36 data (MAE = 2.77 cm) was deemed most suitable for practical applications, enabling timely interventions in cultivation management. This research demonstrates the feasibility and effectiveness of integrating drone imagery, logistic growth curves, and LSTM models for the early and accurate prediction of Kimchi cabbage heights, facilitating data-driven decision-making in precision agriculture for improved crop management and yield optimization.

**Keywords:** drone imagery; long short-term memory; Kimchi cabbage; growth prediction; precision agriculture

## 1. Introduction

South Korea's agricultural sector is grappling with multiple challenges. The aging population and the lingering effects of the COVID-19 pandemic have intensified the existing labor shortage, impacting crucial farm management practices such as crop growth monitoring, fertilization, and harvesting [1]. The country's hilly terrain and diverse cultivation conditions further complicate monitoring efforts, particularly for crops like Kimchi cabbage (*Brassica rapa* subsp. *pekinensis* (Lour.) Hanelt), which are often grown in small-scale, topographically varied environments [2]. Manual field monitoring in such conditions is labor-intensive and prone to inconsistencies.

The South Korean government's Digital New Deal initiative seeks to address these challenges by promoting digital agriculture [3]. This approach harnesses data, networks, and artificial intelligence (AI) to boost agricultural productivity and precision [4,5]. Digital agriculture holds the promise of increased farm income, automation, and sustainability. However, there remains a significant gap in technologies for open-field digital farming, with most advancements currently concentrated in controlled environments like greenhouses [6,7].

Kimchi cabbage, a staple vegetable with seasonal variations, is crucial for South Korea's food security and consumer prices. Autumn cabbage, accounting for the highest production (56.4% in 2020), is particularly vulnerable to price fluctuations due to production variability and climate-related weather events [8]. The labor shortage further compounds these

challenges, leading to increased production costs, inconsistent quality, and unpredictable price swings.

Advancements in digital technology offer potential solutions. AI-based digital farming technologies are being integrated with Kimchi cabbage growth information to improve the production prediction accuracy and enable more scientific farming practices [9]. Traditionally, Kimchi cabbage heights, a key growth indicator, are measured manually in the field. This method is labor-intensive, weather-dependent, and susceptible to variations in accuracy due to the investigator's skill level [10].

Remote sensing (RS) technology presents a valuable alternative. While satellite RS has limitations, due to weather conditions and fixed orbital cycles, recent developments in drone and sensor technology offer a more flexible solution. Drones can capture high-resolution images anytime and anywhere, overcoming the constraints of satellite RS [11,12]. This enables the acquisition of detailed crop and canopy features using image processing techniques. Drone RS allows for rapid, periodic field information collection, facilitating scientific monitoring in conjunction with various analysis technologies [13,14]. The extracted data, such as time series images of Kimchi cabbage growth, can then be used to develop predictive models for Kimchi cabbage growth.

Traditional methods for Kimchi cabbage phenotyping have limitations. While chlorophyll fluorescence imaging has been used for Kimchi cabbage growth modeling, it primarily focuses on leaf metabolism rather than a pixel-level growth prediction, which is crucial for accurately capturing growth patterns over time [15].

Recurrent neural networks (RNNs) are a class of artificial neural networks (ANNs) designed for analyzing sequential data, making them well suited for time series analyses due to their cyclical architecture [16,17]. Their structure allows them to retain information from previous processing steps, making them ideal for tasks like a time series analysis. However, traditional RNNs struggle with long-term dependencies, where patterns emerge over extended sequences [18].

Among the available AI algorithms, long short-term memory (LSTM), a type of RNN, is particularly suitable for analyzing time series data like soil moisture [19], canopy cover [20], and crop height [21,22]. LSTM's structure addresses the vanishing gradient problem in RNNs, enabling it to learn long-term dependencies within time series data.

Kimchi cabbage, like other crops, undergoes distinct phenological stages during its growth cycle. These stages, which include germination, seedling establishment, rosette formation, head development, and maturity, are characterized by specific morphological and physiological changes in the plant [23,24]. Understanding these stages is crucial for interpreting growth patterns and developing accurate prediction models. The timing and duration of each stage can be influenced by various factors, including the genotype, environmental conditions, and management practices. These variations in phenological development can impact the plant's response to its environment and ultimately affect its final height and yield. Therefore, considering the phenological stages of Kimchi cabbage is essential for developing a robust and reliable growth prediction model.

This study aims to develop a technology for the early prediction of Kimchi cabbage heights using drone images and an LSTM-based prediction model. This approach has the potential to address the limitations of current methods and contribute to more efficient and accurate farm management practices, particularly for Kimchi cabbage cultivation.

## 2. Materials and Methods

### 2.1. Study Area

This study was conducted on the autumn Kimchi cabbage testbed at the National Institute of Agricultural Sciences (NAS) in Iseo-myeon, Wanju-gun, Jeonbuk, Republic of Korea (127°02′49.65″ E, 35°49′28.52″ N). The 624 m$^2$ testbed is divided into Zone A (comprising loam) and Zone B (comprising sandy loam), representing the two main soil types used for Kimchi cabbage cultivation in the Republic of Korea (Figure 1). Sandy loam, with larger pores than loam, offers greater permeability. The experiment aimed to

investigate the impact of these distinct soil types on Kimchi cabbage growth, while also utilizing the data from both zones to enhance the versatility of the production model, even though zone-specific modeling showed promising results.

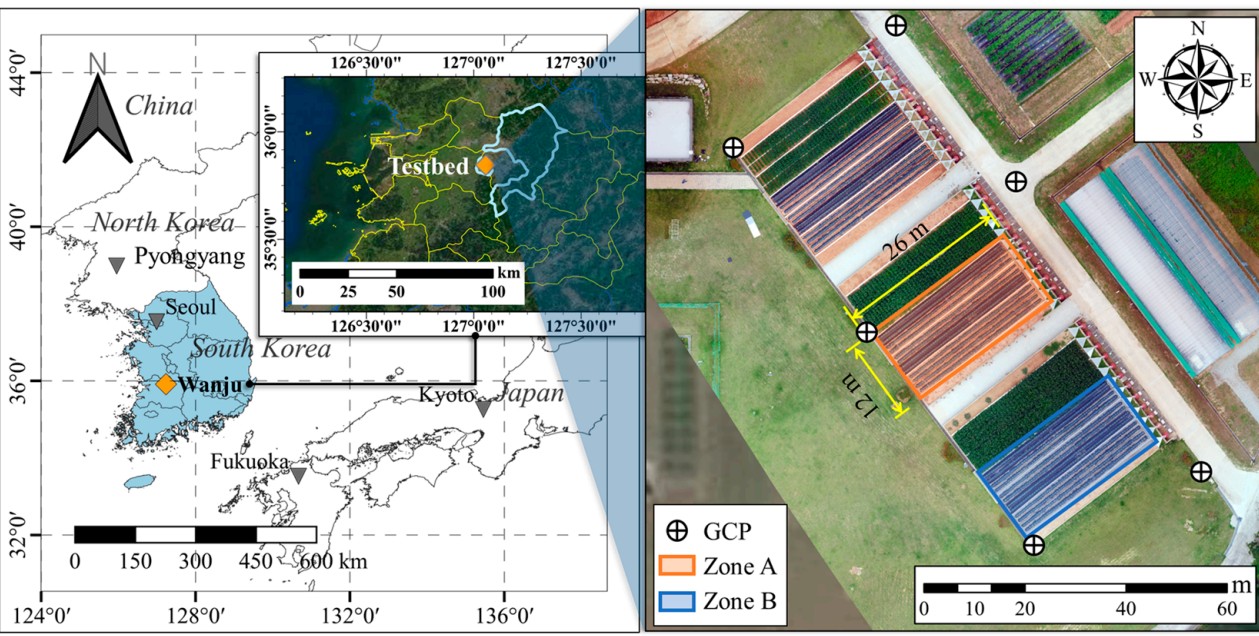

**Figure 1.** Study site overview: Location of the Kimchi cabbage testbed within the Republic of Korea (left) and detailed layout of the testbed at the National Institute of Agricultural Sciences (NAS) showing soil zones (A and B) and ground control point (GCP) locations (right).

On 29 August 2019, 1332 autumn Kimchi cabbages were planted, with 672 in Zone A and 660 in Zone B. The planting followed a 1 m spacing between ridges and 40 cm × 85 cm between individual plants. An automatic drip irrigation system, adjusted according to rainfall, was employed throughout the cultivation period. Six permanent ground control points (GCPs) were selected from concrete structures within the testbed for geo-referencing purposes (Figure 1). The coordinates from these GCPs were then used to perform a geometric correction on the time series drone images captured during this study.

## 2.2. Workflow for the Early Prediction of Kimchi Cabbage Heights Using Drone Imagery and the LSTM Model

This study employed a five-stage workflow (Figure 2) to develop a data-driven model for the early prediction of Kimchi cabbage heights using drone imagery. In the first stage (data acquisition), drone images were captured throughout the growing season, while the corresponding field measurements of the Kimchi cabbage heights were collected. Stage two (image processing) involved generating digital surface model (DSM) and canopy height model (CHM) images from the acquired drone data. Leveraging object-based extraction on the CHM images in stage three, individual Kimchi cabbages were isolated, and the relevant data were extracted. Stage four focused on the model build and testing. Here, growth curves were generated from the extracted data to represent the temporal changes in the Kimchi cabbage height. Subsequently, LSTM models were built for the Kimchi cabbage height prediction. Finally, the accuracy of the models was evaluated, and the predicted heights were visualized for comparison with the actual measurements.

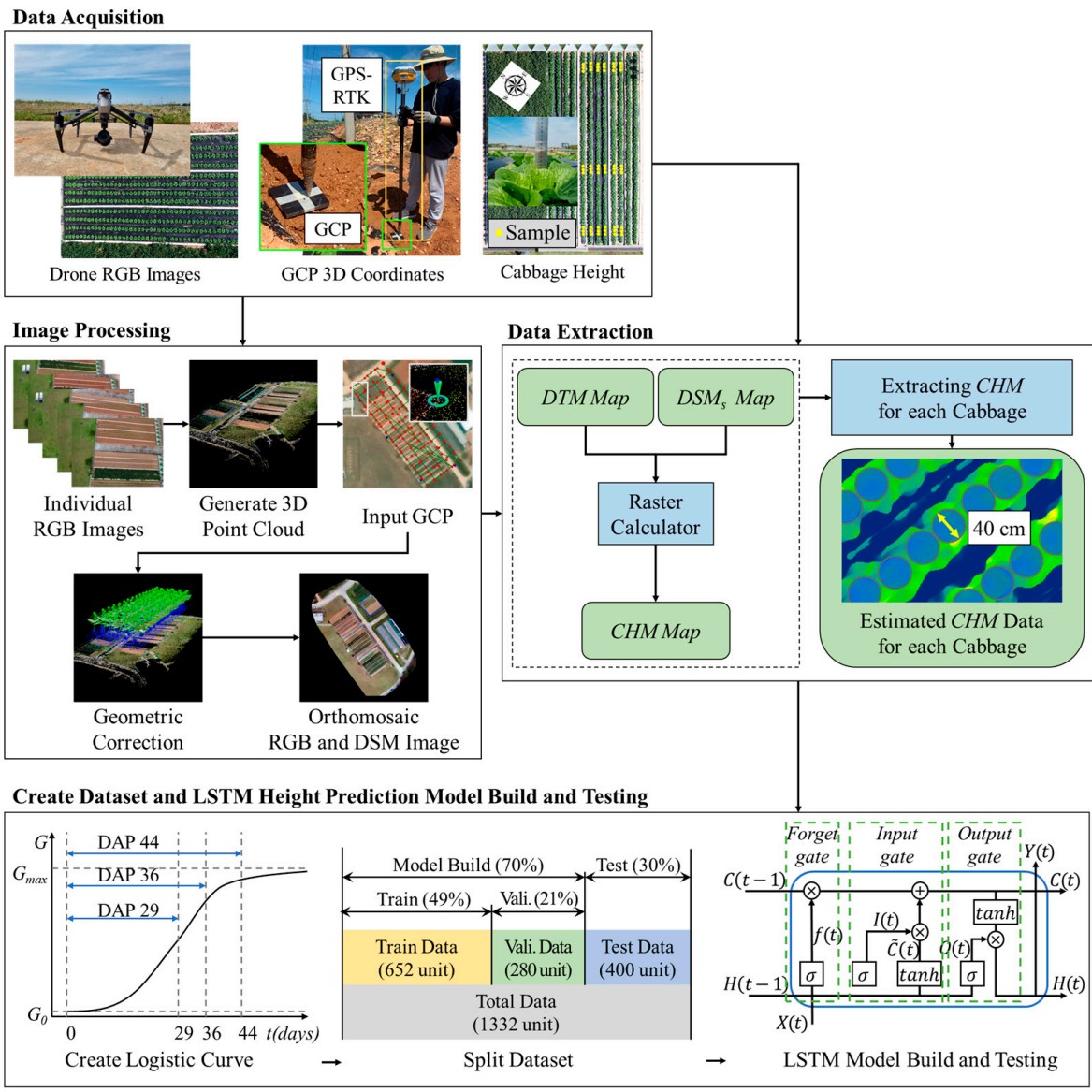

**Figure 2.** Workflow for early prediction of Kimchi cabbage heights using drone imagery and LSTM-based modeling.

### 2.3. Comprehensive Data Acquisition Strategy for Kimchi Cabbage Height Monitoring Using Drone Imagery

A meticulous data acquisition strategy was employed to monitor the Kimchi cabbage growth using drone imagery. High-resolution images of the Kimchi cabbage field were captured by Inspire2 (DJI, Shenzhen, China) drones equipped with Zenmuse-X5S (DJI, Shenzhen, China) RGB sensors (Table 1). The flight parameters, including flight path, altitude, overlap, and shooting angle, were carefully optimized, considering factors such as the sensor capabilities, desired image resolution, drone limitations, and flight time. To capture the temporal dynamics of the plant growth, the images were acquired at approximately one-week intervals, beginning on 28 August 2019, one day before planting, to establish a baseline digital terrain model (DTM) representing the bare field. A total of 13 image acquisition missions were conducted until the harvest date of 8 November 2019 (Table 2).

**Table 1.** S Drone and sensor specifications for Kimchi cabbage field image acquisition.

| Material | Item | Specification |
|---|---|---|
| Inspire2 | Manufacturer | DJI, Shenzhen, China |
| | Max take-off weight | 4000 g |
| | Max flight time | Approx. 27 min |
| Zenmuse-X5S | Manufacturer | DJI, Shenzhen, China |
| | Weight | 461 g |
| | Sensor | CMOS, 4/3″; effective pixels: 20.8 MP |
| | FOV | 72° (with DJI MFT 15 mm/1.7ASPH) |
| | Photo resolutions | 4:3, 5280 × 3956 pixel |
| | | 16:9, 5280 × 2970 pixel |

Sources: https://www.dji.com/kr/inspire-2/ accessed on 14 July 2024. and https://www.dji.com/kr/zenmuse-x5s/ accessed on 14 July 2024.

**Table 2.** Drone and field data collection for Kimchi cabbage growth monitoring.

| Date | Days After Planting (DAPs) | Drone Imagery | Field Survey Data | Planting Vegetation |
|---|---|---|---|---|
| 28 August 2019 | −1 | O | X | X |
| 29 August 2019 | 0 | O | X | O |
| 31 August 2019 | 2 | O | O | O |
| 10 September 2019 | 12 | O | O | O |
| 17 September 2019 | 19 | O | O | O |
| 20 September 2019 | 22 | O | O | O |
| 27 September 2019 | 29 | O | O | O |
| 4 October 2019 | 36 | O | O | O |
| 12 October 2019 | 44 | O | O | O |
| 18 October 2019 | 50 | O | O | O |
| 25 October 2019 | 57 | O | O | O |
| 1 November 2019 | 64 | O | O | O |
| 8 November 2019 | 71 | O | O | O |

The autonomous drone flights were facilitated by Pix4Dcapture version 4.13.1 software (Pix4D, Prilly, Switzerland), allowing precise control over the flight parameters. To enable the detailed analysis of the individual Kimchi cabbage plants, the drone image acquisition altitude was set to 30 m. This resulted in a ground sample distance (GSD) of approximately 0.7 cm/pixel. The choice of such a high-resolution GSD was driven by the need to accurately capture the individual plant heights in the dense planting conditions of the Kimchi cabbage field. While a lower altitude and smaller GSD could potentially increase the accuracy of the height estimation, it would also lead to an increased flight time and data processing requirements. The selected GSD of 0.7 cm/pixel struck a balance between accuracy and practicality, allowing for detailed plant-level analysis while maintaining a reasonable flight time and data processing workload [14]. The importance of capturing fine-grained spatial information for precision agriculture applications, such as identifying growth variations and detecting potential issues at the individual plant level, further justifies the use of high-resolution imagery in this study.

The total area covered by the drone imagery was 4752 m$^2$. A double grid flight path with a 75° acquisition angle was chosen to optimize the accuracy of the Z-axis (height) data, ensuring detailed 3D information capture of the Kimchi cabbages [25]. Each image capture cycle took approximately 10 min, with a longitudinal and lateral overlap of 75% and a flight speed of 3 m/s (10.8 km/h).

The acquired RGB images and cloud data were processed into DSM orthomosaics and RGB orthomosaics, which were then geometrically corrected using Pix4Dmapper (https://www.pix4d.com/product/pix4dmapper-photogrammetry-software, Pix4D, Prilly, Switzerland) software. The geometric correction ensured the accurate alignment of the acquired images with reference locations, enabling the spatial analysis. The GCPs, predefined

ground locations with known coordinates established using permanent structures within the field, were used for this correction [25]. By matching an image of the GCP locations with their corresponding coordinates for each acquisition date, the image position was effectively calibrated over time, facilitating a comparative analysis of the Kimchi cabbage growth at the same location across the images. Both the GCP coordinates and the image coordinate system were unified as WGS 84 UTM zone 52 N (Figure 3).

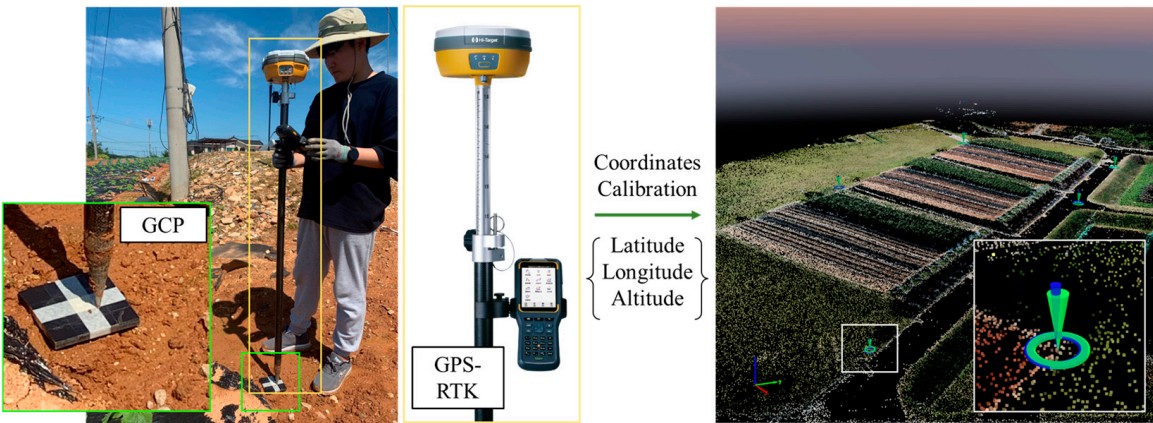

**Figure 3.** Geometric correction and coordinate system unification of drone imagery using GCPs.

Field measurements of the Kimchi cabbage height were conducted on the same dates as the drone image acquisitions, except for 29 August 2019. These measurements were manually taken using a ruler from the top of the gyrus to the top of the cabbage leaf, excluding the roots, for comparison with the CHM. The top of the gyrus was chosen as the starting point for the height measurement because it represents a stable and easily identifiable reference point on the Kimchi cabbage plant. This approach ensures a consistency in measurements across the different growth stages and individual plants, as the gyrus remains relatively fixed while the leaves continue to expand. A total of 56 sample cabbages (18 cabbages per zone), distributed across the upper, middle, and lower zones of Zone B, were selected for the field measurements. These measurements, taken at various time points throughout the growing season, provided the ground truth data for validating the accuracy of the drone-derived height estimates and training the LSTM model for the height prediction.

This comprehensive data acquisition strategy, combining high-resolution drone imagery with carefully planned field measurements, provided a robust dataset for the subsequent analysis of Kimchi cabbage growth using LSTM techniques.

### 2.4. Estimation of Kimchi Cabbage Height Using Canopy Height Models Derived from Drone Imagery

RS techniques can estimate Kimchi cabbage heights by analyzing changes in a DSM generated from drone-captured RGB images. The DSM is created by projecting the 3D point cloud heights, obtained using the structure from motion (SfM) algorithm, onto the corresponding 2D RGB image. Each pixel value in the DSM represents the surface elevation, and its accuracy depends on the resolution and overlap of the captured RGB images [26].

A CHM is generated for each Kimchi cabbage growth stage to estimate the Kimchi cabbage height using the drone RS imagery acquired after planting. The CHM is calculated by subtracting the DTM representing the bare earth surface from the DSM representing the surface including vegetation (Figure 4). The difference between the DSM and DTM provides an estimate of the Kimchi cabbage height (CHM) at each growth stage, as expressed in Equation (1):

$$CHM = DSM_s - DTM \tag{1}$$

where CHM is the Kimchi cabbage height for each growth stage estimated using drone RS, $DSM_s$ is the surface model for each Kimchi cabbage growth stage, and DTM is the unique numerical terrain model of the farmland [27,28].

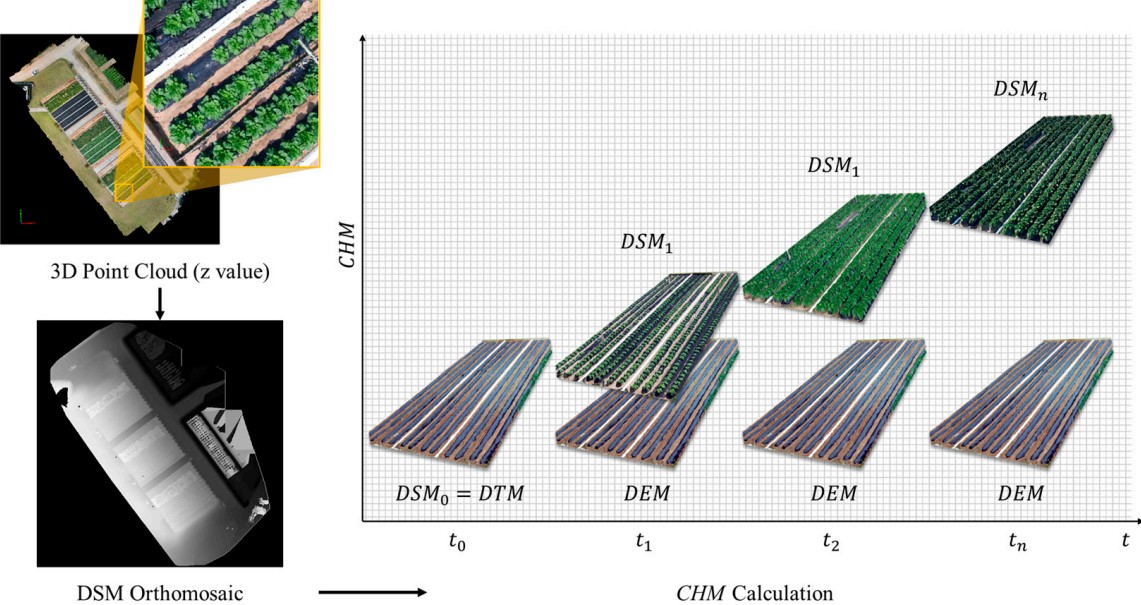

**Figure 4.** Schematic of generating a canopy height model (CHM) from time series digital surface models (DSMs) and a digital terrain model (DTM).

The Kimchi cabbage height for each individual plant was estimated based on a time series analysis of the DSM and CHM images using ArcGIS Pro (https://www.esri.com/en-us/arcgis/products/arcgis-pro/overview, ESRI, Redlands, CA, USA), a comprehensive geospatial information analysis software with functionalities for image processing, 2D/3D analysis, visualization, and data management. The method for extracting the height of the individual cabbage plants involved creating a central point shapefile for each cabbage plant, as shown in Figure 2, "Data Extraction". From each CHM image corresponding to different dates, the highest pixel value within a 40 cm radius of the central point was extracted.

*2.5. Kimchi Cabbage Height Prediction Using Logistic Growth Curve Interpolation of Drone-Derived Data*

The logistic growth curve, introduced by Verhulst (1845), is a well-established mathematical model used to analyze growth processes in various fields, including biology [29–31]. Due to its ability to represent sigmoid growth patterns, this model has been successfully applied to study crop growth trends [32,33]. The logistic growth curve can be parameterized by different functions, and the specific form used in this study is shown in Equation (2).

$$G(t) = \frac{G_{max}}{1 + \left( \frac{G_{max}}{G_0} - 1 \right) \exp(-kt)} \tag{2}$$

where $G(t)$ is the growth value at time t, $G_0$ is the initial growth value, $G_{max}$ is the maximum growth value, and $k$ is a time-related parameter [34,35].

In this study, the logistic growth curve is employed to interpolate the missing Kimchi cabbage height data acquired through drone imagery. The growth curve requires the final height data, and, since the growth of individual cabbage plants varies, using only the initial data to predict the final height growth has limitations. Therefore, this study used an LSTM model to create a predictive model, interpolating the initial missing data into a growth curve to use as input data. This interpolation capability is crucial for obtaining a continuous

growth curve, enabling a more comprehensive analysis of growth patterns and facilitating the training of the LSTM model for the height prediction.

The parameters of the logistic growth curve were estimated using the least squares method based on weekly CHM data to accurately model the growth curve of Kimchi cabbage. To account for variations in the growth conditions of individual cabbage plants in the model, 1332 cabbage growth curves were generated and used as training and testing data. Notably, the Python library Scipy and its optimization algorithms were used for efficient parameter derivation (Figure 5).

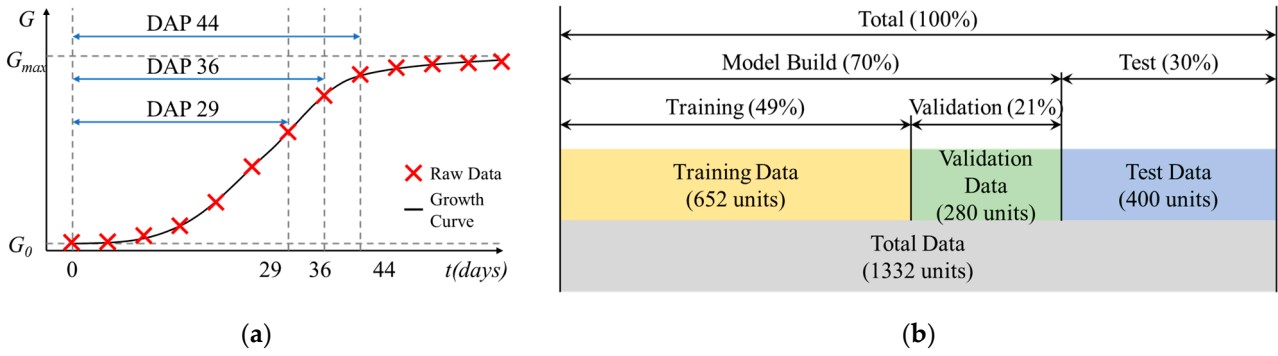

(**a**)  (**b**)

**Figure 5.** Data preparation for LSTM model development: (**a**) logistic growth curve fitting using CHM data at key growth stages; and (**b**) dataset partitioning for training, validation, and testing.

To predict the Kimchi cabbage height at harvest, the Kimchi cabbage growth data for each plant were collected at one-day intervals, specifically at 29, 36, and 44 days after planting (DAPs). The selection of DAPs 29, 36, and 44 was based on two key considerations. First, these dates fell within the pre-heading stage during the growth acceleration phase, a critical period when cabbage height is actively increasing and likely to be indicative of final size. Second, these dates were chosen to coincide with the drone image acquisition dates, ensuring data availability for the model training and validation.

These data points served as the learning data for the logistic growth curve. The resulting growth curve was then used to train a model for predicting the Kimchi cabbage height on the harvest day. In Figure 5a, we can see the logistic curves generated using the data from three different time points (29, 36, and 44 DAPs). These curves were used for the model training. Figure 5b illustrates the data partitioning strategy. The Kimchi cabbage data (1332 data points for each date) were divided into 70% for model training and 30% for testing, without distinction between areas A and B. The training data were further divided into 70% for learning and 30% for verification purposes.

### 2.6. Kimchi Cabbage Height Prediction Using LSTM Network Model with Time-Series Growth Data

LSTM networks address this limitation by incorporating memory cells that can store and manage information over longer periods [24]. As illustrated in Figure 6, LSTMs employ forget gates, input gates, and output gates to regulate the flow of information within the network (Table 3) [36]. These gates determine the retention of past information, the incorporation of new information, and the output for the next cell in the sequence.

An LSTM model was designed to predict Kimchi cabbage heights at harvest using CHM data. The model, built with TensorFlow in Python, utilized a three-layer structure for in-depth learning and improved prediction accuracy. It employed a three-layer structure to facilitate in-depth learning and improve prediction accuracy. The coefficient of determination ($R^2$), mean absolute error (MAE), and root mean square error (RMSE) loss functions were used for the model compilation. To ensure consistency in evaluating the models trained with data from different time intervals (29, 36, and 44 DAPs), the number of training iterations was fixed at 3000 for all the models.

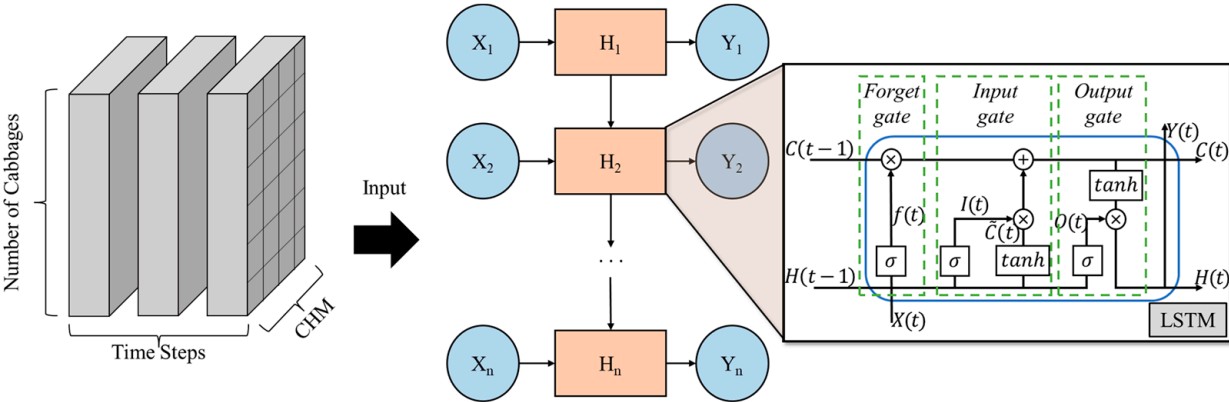

**Figure 6.** LSTM network architecture for Kimchi cabbage height prediction, illustrating the input data structure, hidden states, and the internal gating mechanism of an LSTM cell. The input data are a three-dimensional matrix with dimensions representing the number of Kimchi cabbages, time steps, and CHM for each day.

**Table 3.** LSTM model architecture: gates, activation functions, and parameters for Kimchi cabbage height prediction.

| | Layer | Equation | Parameter |
|---|---|---|---|
| LSTM Layer | Forget gate | $f(t) = sigmoid\left(U_f X(t) + W_f H(t-1) + b_f\right.$ | $U_f, W_f, b_f$ |
| | Input gate | $\widetilde{C}(t) = tanh(U_c X(t) + W_c H(t-1) + b_c$ <br> $I(t) = sigmoid(U_i X(t) + W_i H(t-1) + b_i$ | $U_c, W_c, b_c$ <br> $U_i, W_i, b_i$ |
| | Cell state | $C(t) = F(t)C(t-1) + I(t)\widetilde{C}(t)$ | - |
| | Output gate | $H(t) = O(t)tanh(C(t))$ <br> $O(t) = sigmoid(U_o X(t) + W_o H(t-1) + b_o$ | $U_o, W_o, b_o$ |

The LSTM model received the CHM data, extracted from the logistic growth curve, as the input. These data were formatted as a three-dimensional matrix encompassing Kimchi cabbages (number of data points), time steps (time intervals), and the CHM (independent variables) (Figure 6). The model was then used to predict the CHM for each Kimchi cabbage on the harvest day (8 November 2019).

The accuracy of the predicted Kimchi cabbage height values was assessed using $R^2$, MAE, and RMSE. These metrics were compared with the actual CHM data obtained from drone imagery on November 8th. Finally, the predicted CHM was mapped to visualize the spatial distribution of growth across the field.

## 3. Results

### 3.1. Monitoring Kimchi Cabbage Growth Using Drone Imagery and Derived Height Models

Drone imagery was successfully acquired 13 times between 28 August 2019 (one day before planting) and 8 November 2019 (harvest day) for the Kimchi cabbage testbeds in NAS, Republic of Korea. Each image acquisition captured 312 images with a GSD of approximately 0.7 cm, enabling the high-resolution analysis of individual Kimchi cabbages. The accuracy of the imagery, based on the GCP coordinates, showed an average RMSE error for each image ranging from 0.0052 m to 0.0154 m along the x-, y-, and z-axes. The error along the z-axis had an overall average RMSE of 0.0108 m, allowing for the analysis of the cabbage height in centimeters.

Figure 7 demonstrates the effectiveness of drone imagery for monitoring Kimchi cabbage growth. The figure displays (a) RGB orthomosaics, (b) DSMs, and (c) CHM maps for three representative dates (DAP 0, DAP 36, and DAP 71). A visual inspection of the RGB orthomosaics (Figure 7a) allows for a general assessment of the plant health based

on color variations. However, this method cannot quantify the Kimchi cabbage height. DSMs (Figure 7b) address this limitation by representing the Kimchi cabbage height as the ellipsoid height within a reference coordinate system. While DSMs cannot provide the absolute height of individual plants, they enable the visualization of height variations across the field over time through a time series analysis. CHM maps (Figure 7c) offer the most precise height information. These maps are generated by subtracting the DTM from the DSM orthoimage. This numerical representation removes the influence of the terrain slope, allowing for the accurate extraction of Kimchi cabbage height data. As evident in Figure 7c, the CHM map for DAP 71 reveals slower growth in Zone A and the complete absence of Kimchi cabbages in Zone B, likely due to environmental factors.

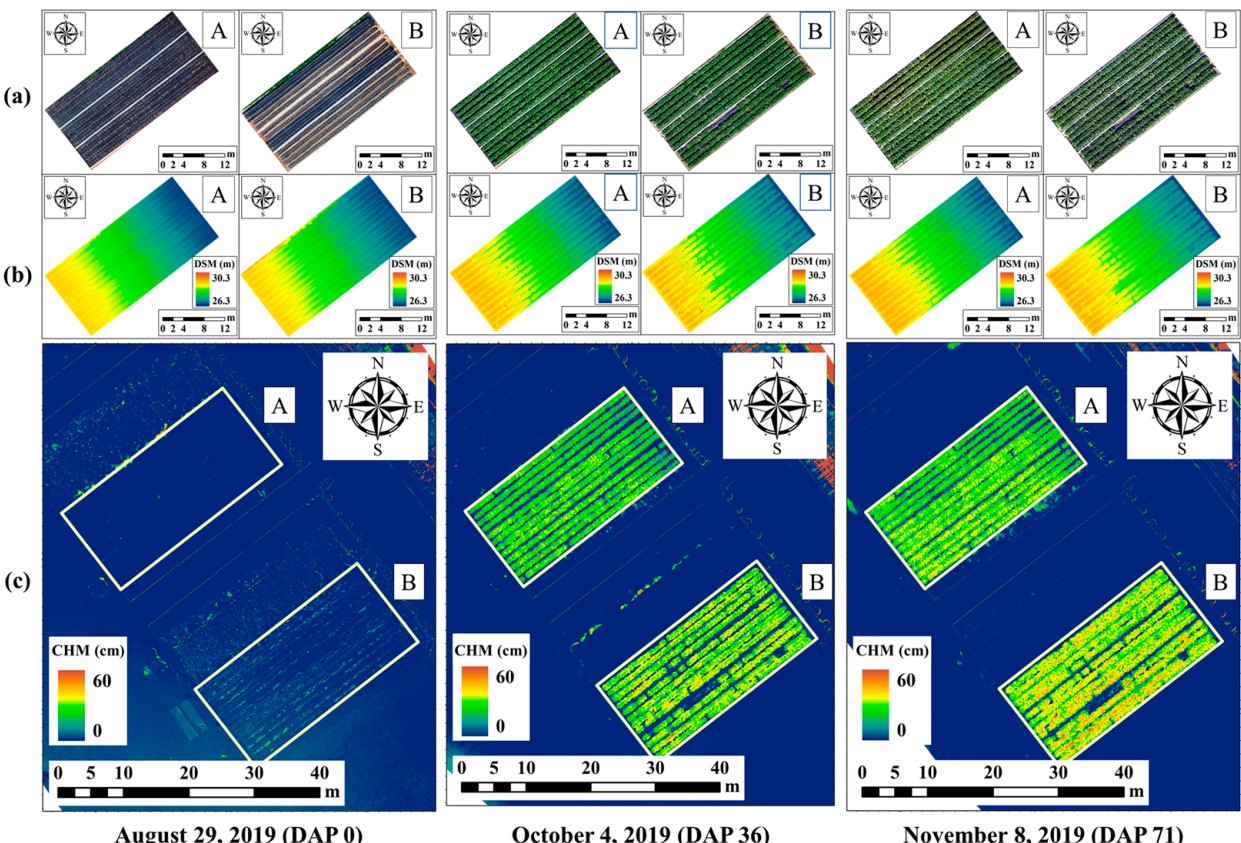

**Figure 7.** Temporal visualization of Kimchi cabbage growth in Zone A (loam) and Zone B (sandy loam) using drone imagery: (**a**) RGB orthomosaics, (**b**) DSMs, and (**c**) CHMs at three key growth stages (DAPs 0, 36, and 71).

Drone imagery combined with DSM and CHM calculations provides a valuable tool for monitoring Kimchi cabbage growth. CHM maps, in particular, offer a quantitative and spatially explicit assessment of Kimchi cabbage heights throughout the growing season.

### *3.2. Accuracy Assessment of Kimchi Cabbage Height Estimation from Drone Imagery*

This section evaluates the accuracy of the Kimchi cabbage height measurements derived from drone imagery. Field surveys were conducted to collect reference data on the actual measurement of the cabbage height for 56 representative Kimchi cabbages. The Kimchi cabbage heights were measured on several dates throughout the growing season, starting from planting on 29 August 2019. However, some Kimchi cabbages were identified as vacant hills during the 10 September survey and excluded from the subsequent measurements, resulting in a final sample size of 50 Kimchi cabbages.

Figure 8a depicts the distribution of the field measurements of the Kimchi cabbage height values using a boxplot. The boxplot reveals a growth range of approximately 0 to

50 cm, with the most significant growth occurring between 10 September and 4 October. Additionally, the spread of the Kimchi cabbage height increased substantially by October 4th, reaching a difference of about 25 cm between the maximum and minimum values. This spread then narrowed after 12 October.

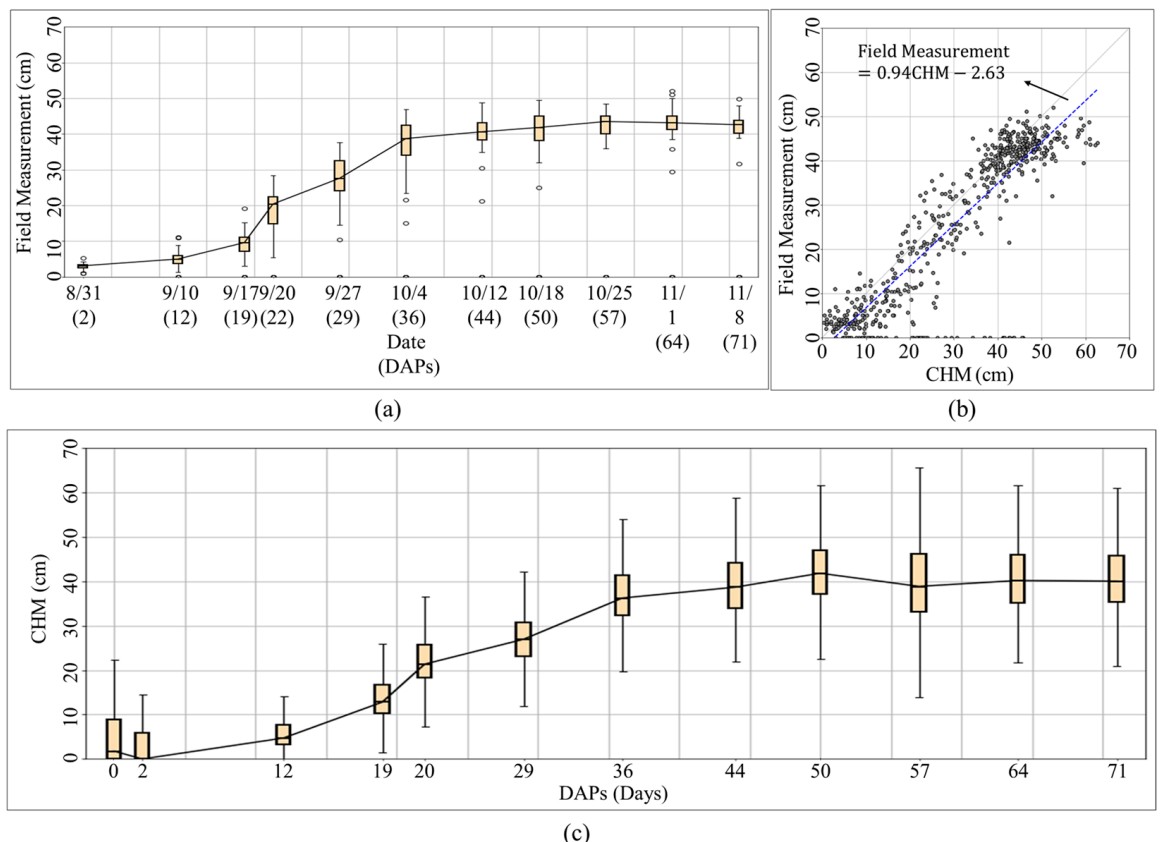

**Figure 8.** Validation of Kimchi cabbage height estimation from drone imagery: (**a**) boxplot of field-measured plant heights, (**b**) scatter plot comparing field-measured heights with canopy heights derived from CHMs, and (**c**) temporal progression of CHM in Zones A and B.

To assess the accuracy of the drone-based height measurements (for the CHM), the heights of 50 sample cabbage plants obtained from field surveys were compared with the corresponding CHM data. This comparison is visualized in the scatter plot of Figure 8b. A linear trendline was fitted to the data points using a least squares regression analysis. The slope of the trendline was approximately 0.94, indicating a slight overestimation of the Kimchi cabbage height by the CHM values, particularly for the taller Kimchi cabbages. The y-intercept of −2.63 further suggests a systematic bias towards higher values in the CHM data. While most vacant hill Kimchi cabbages were correctly identified as having low heights, some outliers with estimated heights exceeding 30 cm were observed. Figure 8c presents a growth curve for all the Kimchi cabbages in Zones A and B, estimated using drone imagery throughout the 2019 growing season. The CHM exhibited slow growth until DAP 12, followed by a period of rapid growth until DAP 44. From DAP 50 onwards, coinciding with the maturation and harvesting stages, the growth curve plateaus or shows minimal variation. This pattern aligns with the biological transition from vegetative growth in the pre-determinant stage to reproductive growth upon reaching the determinant stage, where Kimchi cabbage heads begin to form and solidify.

Table 4 summarizes the accuracy evaluation using the MAE and RMSE. The MAE provides a straightforward interpretation of the average error between the CHM and the field measurement height values. The RMSE assigns higher weights to larger errors, providing insights into the distribution of errors. Overall, the MAE values varied across

different growth stages but remained relatively low, averaging 4.21 cm throughout the entire monitoring period. The CHM values exhibited an overall error of 3–6 cm compared to the measured heights, with a maximum error of less than 10% of the typical harvest-time Kimchi cabbage height (40–60 cm). While the RMSE indicated the presence of some larger errors in individual Kimchi cabbage height measurements, the overall error remained acceptable for practical applications. These findings demonstrate that the Kimchi cabbage height estimates derived from drone imagery and the CHM calculation method provide a valuable tool for monitoring the Kimchi cabbage growth status.

**Table 4.** Error metrics (MAE and RMSE) for drone-based Kimchi cabbage height estimation at various DAPs.

| Date (DAPs) | MAE (cm) | RMSE (cm) |
| --- | --- | --- |
| 08/31 (2) | 4.58 | 5.73 |
| 09/10 (12) | 3.98 | 5.09 |
| 09/17 (19) | 6.29 | 7.68 |
| 09/20 (22) | 4.25 | 5.29 |
| 09/27 (29) | 4.53 | 5.59 |
| 10/04 (36) | 5.10 | 6.36 |
| 10/12 (44) | 4.15 | 5.75 |
| 10/18 (50) | 4.94 | 6.51 |
| 10/25 (57) | 4.81 | 6.22 |
| 11/01 (64) | 3.79 | 5.29 |
| 11/08 (71) | 3.78 | 5.42 |
| Total | 4.21 | 5.52 |

### 3.3. Modeling Kimchi Cabbage Growth Dynamics Using Logistic Growth Curves

A logistic growth curve was employed to interpolate the missing values in the Kimchi cabbage height data and to model the overall growth trend (Figure 8c). Figure 9 illustrates logistic growth curves for three scenarios, each representing Kimchi cabbage growth up to a specific time point: 29, 36, and 44 DAPs. Additionally, a boxplot in Figure 9 visualizes the daily variations in the CHM, the estimated plant height from drone imagery.

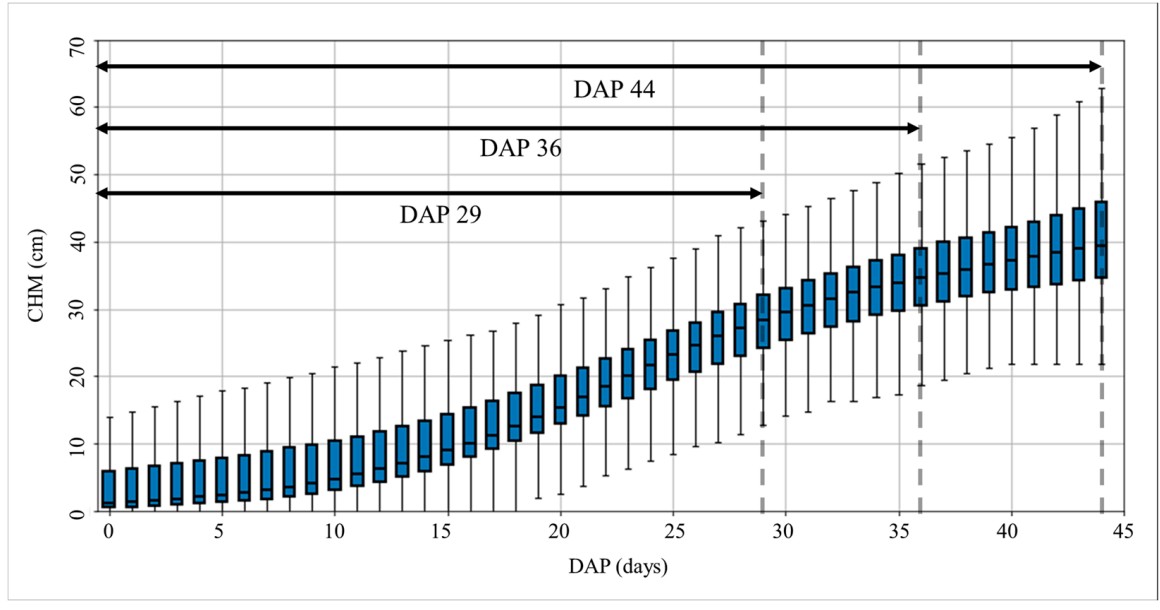

**Figure 9.** Kimchi cabbage growth dynamics: boxplots depicting daily canopy height (CHM) variations alongside logistic growth curves fitted to CHM data at key growth stages (DAPs 29, 36, and 44).

The logistic growth curves effectively demonstrate the model's capability to capture the dynamic growth patterns of Kimchi cabbage throughout the growing season. The curve for DAP 29 reflects the rapid initial growth phase, while the curve for DAP 36 extends further, capturing changes into the later, slower growth stage. The DAP 44 curve encompasses variations across the entire growth period, including the late, slow growth stage.

Figure 8c complements Figure 9 by presenting the growth curve for all the Kimchi cabbages in Zones A and B throughout the 2019 growing season. This figure reveals a slow initial growth phase (until DAP 12), followed by rapid growth (until DAP 44), and finally a plateau or minimal variation from DAP 50 onwards, coinciding with maturation and harvesting. This pattern aligns with the biological transition from vegetative to reproductive growth, where Kimchi cabbage heads form and solidify. Together, Figures 8c and 9 provide a comprehensive visualization of the Kimchi cabbage growth dynamics throughout the growing season, captured through both raw data and the logistic growth model.

*3.4. LSTM-Based Kimchi Cabbage Height Prediction for Cultivation Management: Model Training, Evaluation, and Selection*

The LSTM model was trained using the logistic growth curves generated in Figure 9, representing the predicted Kimchi cabbage height at various DAPs. For each Kimchi cabbage plant, the CHM values at DAPs 29, 36, and 44 were used as the independent variables, while the CHM at DAP 71 (harvest) served as the dependent variable. The model training was conducted on a workstation equipped with an Intel Core i9-10940X CPU and an NVIDIA GeForce RTX 3080 Ti GPU. Each iteration of the training process took approximately 30 milliseconds, with a total of 3000 iterations completed in 30 s. The model output was divided into verification and testing datasets (30% of the data) to assess the learning accuracy and predictive power on unseen data, respectively.

Table 5 summarizes the learning results for the three models (DAPs 29, 36, and 44). All the models exhibited a good consistency, with low $R^2$ differences (<0.04) between the verification and testing and minimal differences between the MAE and RMSE (within 0.5 cm), suggesting no significant overfitting. The test results revealed an increasing prediction accuracy with longer time steps for the learning, with the DAP 44 model achieving the highest accuracy (MAE of 2.23 cm), followed by DAP 36 (MAE of 2.77 cm) and DAP 29 (MAE of 4.35 cm).

**Table 5.** Predictive performance of LSTM models for Kimchi cabbage height at harvest (DAP 71): evaluation using training data from different growth stages (DAPs 29, 36, and 44).

| Index | DAP 29 | | | DAP 36 | | | DAP 44 | | |
|---|---|---|---|---|---|---|---|---|---|
| | Vali. * | Test | Total | Vali. | Test | Total | Vali. | Test | Total |
| $R^2$ | 0.48 | 0.45 | 0.46 | 0.76 | 0.74 | 0.77 | 0.85 | 0.85 | 0.83 |
| MAE (cm) | 4.12 | 4.35 | 4.22 | 2.84 | 2.77 | 2.77 | 2.54 | 2.23 | 2.42 |
| RMSE (cm) | 5.74 | 5.75 | 5.75 | 3.88 | 3.83 | 3.74 | 3.31 | 2.92 | 3.26 |

* Vali.: Validation.

The lower $R^2$ value for the DAP 29 model can be attributed to the limited training data available up to 29 days after planting. This model, trained on early-stage growth data, may not fully capture the complex growth patterns and variations that occur in later stages, leading to a reduced prediction accuracy.

Figure 10 presents the predicted Kimchi cabbage height maps generated by applying the three models to the entire field. The results indicate that the predicted heights in Zone A (loam) primarily ranged from 30 to 40 cm, while Zone B (sandy loam) exhibited heights between 40 and 55 cm. This 10–15 cm difference aligns with observed soil condition variations and the slightly faster growth rates reported for Kimchi cabbage in loam compared to sandy loam [10]. This trend is further reflected in the smaller height discrepancies between the measured and predicted values in sandy loam (Zone B) during the early growth stages (Figure 10a).

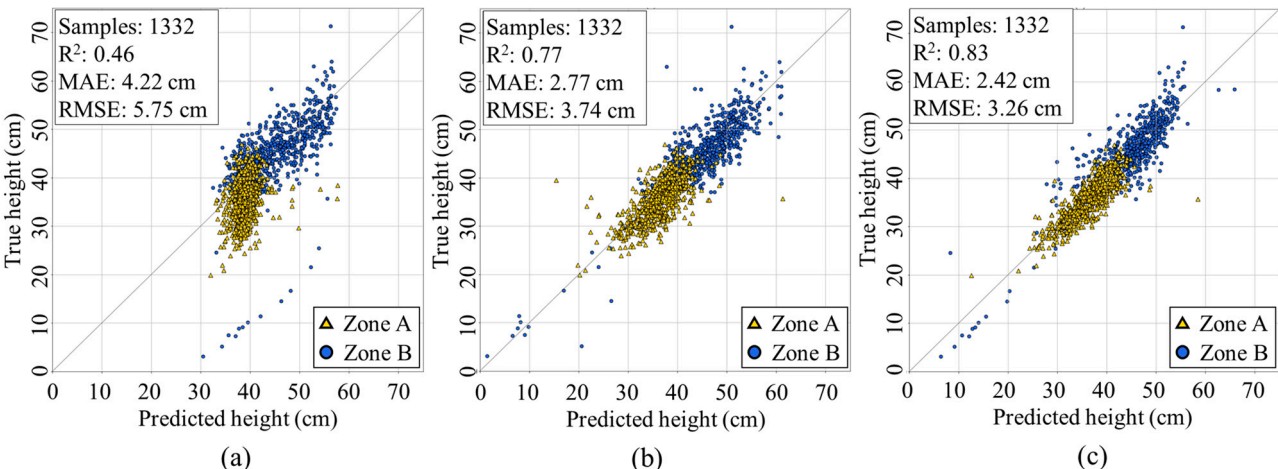

**Figure 10.** Evaluation of LSTM models for Kimchi cabbage height prediction at different training stages: (**a**) DAP 29, (**b**) DAP 36, and (**c**) DAP 44. Scatter plots compare predicted heights against true heights for both Zone A (loam) and Zone B (sandy loam), with corresponding R², MAE, and RMSE values.

The deviance from the line observed in Figure 10a for the LSTM model of DAP 29 in Zone A might be attributed to localized variations in soil properties, the microclimate, or potential inconsistencies in data collection or processing at that specific location. These factors could have influenced the growth patterns of the Kimchi cabbage in Zone A, leading to a discrepancy between the predicted and measured heights.

While the DAP 44 model offered the highest accuracy, its timing near the end of growth limits its applicability for cultivation management interventions. The DAP 36 model, despite a slightly lower accuracy, achieved reasonable prediction errors (MAE 2.77 cm and RMSE 3.74 cm for DAP 71) and coincides with ongoing vegetative growth, allowing for timely adjustments to the agricultural plan. Therefore, considering the trade-off between the accuracy and growth stage, the DAP 36 model was deemed most suitable for informing cultivation management decisions.

*3.5. Visualizing Kimchi Cabbage Height Predictions with Spatiotemporal Maps for Targeted Crop Management*

The predicted final height of each Kimchi cabbage obtained from the LSTM model was used to generate color-coded maps. Each map divided the predicted heights into 10 cm intervals, with distinct colors representing different height ranges (Figure 11).

Figure 11 presents a reference map for comparison, constructed based on the DAP 71 (harvest day) predictions from each model and a field image captured on 8 November 2019. Zone A (loam soil) and Zone B (sandy loam) in the reference map exhibit distinct characteristics due to the soil composition. Zone A primarily shows Kimchi cabbage heights ranging from 30 to 50 cm, while Zone B displays a higher concentration of Kimchi cabbages exceeding 40 cm. Additionally, the reference map includes features marked by arrows indicating a general trend of decreasing Kimchi cabbage heights in Zone A and locations with frequent vacant hills (Figure 11 (i),(ii)).

The Kimchi cabbage height prediction maps for DAP 29, DAP 36, and DAP 44 were compared visually with the reference map (Figure 11). Here is a detailed breakdown of the comparisons:

DAP 29: The model captured the overall height distribution in Zones A and B, with Zone B showing higher values. However, it failed to accurately reflect the decreasing height trend in Zone A (i) and the vacant hill characteristics (ii), predicting heights of 20–50 cm which deviated from the measured values. Zone B displayed a generally similar pattern, although the lower right sections were underestimated.

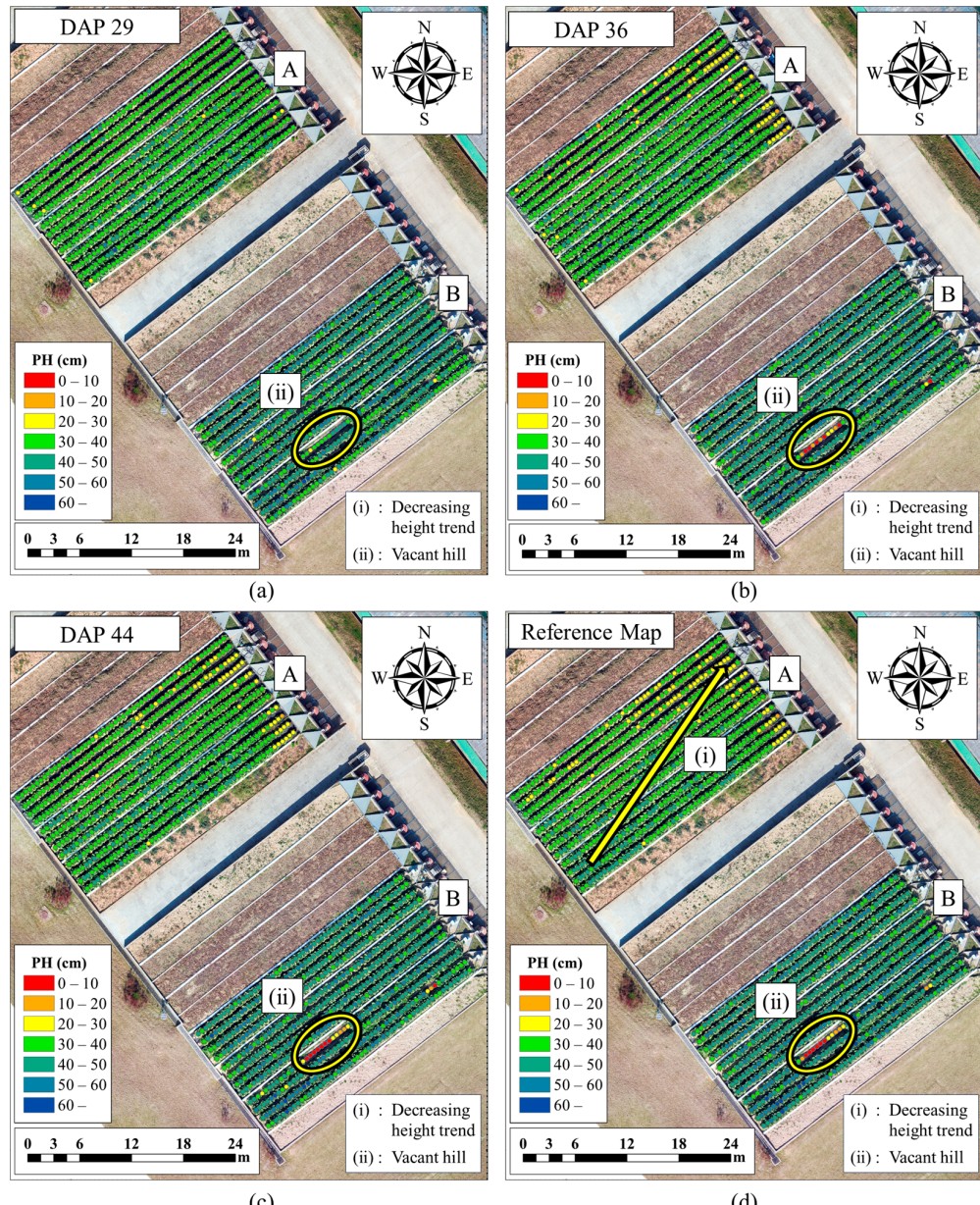

**Figure 11.** Spatiotemporal visualization of predicted Kimchi cabbage height in Zone A (loam) and Zone B (sandy loam): comparison of LSTM model-generated height maps at different growth stages (**a**) DAP 29, (**b**) DAP 36, and (**c**) DAP 44 with a (**d**) reference map based on field measurements at harvest (DAP 71).

DAP 36: This model achieved a more accurate prediction, particularly in the northern region of Zone A, where the Kimchi cabbage heights of 20–40 cm aligned well with the reference map. The vacant hill area (ii) also showed similar values and distribution trends. Overall, the DAP 36 model was considered suitable for Kimchi cabbage height prediction, coinciding with a period of moderate Kimchi cabbage growth, which is valuable for informed growth prediction, prescription, and management practices.

DAP 44: While exhibiting the most similar overall distribution tendency to the reference map in both zones, the DAP 44 model yielded slightly higher predictions in the northern part of Zone A and slightly lower predictions in the southern part. Zone B was also generally underestimated. Despite these discrepancies, both the DAP 36 and DAP 44 models successfully captured the key features observed in the reference map.

By visually analyzing the prediction maps, it was evident that the DAP 36 model results most closely resembled the measured values. When combined with the error evaluation from the previous section, the DAP 36 model emerged as the preferable choice for informing agricultural planning decisions. This preference stems from the DAP 36 model's ability to predict Kimchi cabbage heights during a critical growth stage, allowing for timely interventions through fertilization or chemical treatments if necessary. While the DAP 44 model offered a slightly higher accuracy, the timing of its prediction (closer to harvest) limits its applicability for growth-related management practices.

In conclusion, the Kimchi cabbage height prediction maps generated from the LSTM model provide a valuable tool for visualizing and analyzing spatial variations in Kimchi cabbage growth. The DAP 36 model, in particular, offers a practical solution for precision agriculture applications by enabling informed decision-making during the crucial growth stages of Kimchi cabbage cultivation.

## 4. Discussion

### 4.1. Limitations of RGB Orthoimages and CHM Data for Kimchi Cabbage Height Estimation

RGB orthoimages (Figure 7a) and DSMs (Figure 7a) offer valuable visual insights into crop growth patterns over time, but their inherent limitations prevent the direct translation of color information or relative height differences into precise quantitative height measurements. This is because RGB orthoimages primarily capture spectral reflectance, while DSMs represent relative elevation variations rather than absolute heights. Consequently, while these data sources provide valuable qualitative observations of growth trends and potential anomalies, they lack the quantitative precision necessary for accurate height estimations and growth modeling.

To overcome these limitations, we employed the CHM calculation method, which derives numerical height information by subtracting the DTM representing the bare earth surface from the DSM representing the terrain with vegetation. This approach, consistent with previous studies utilizing CHMs for crop height estimations [26,27], effectively mitigates the influence of terrain variations, enabling a more accurate estimation of Kimchi cabbage heights at each growth stage. However, it is crucial to acknowledge that the accuracy of CHM-based height estimations can be influenced by factors such as image resolution, terrain complexity, and vegetation density [28]. These factors can introduce errors in the DSM and DTM, which subsequently propagate to the CHM and affect the accuracy of the height measurements. Therefore, while CHMs offer a valuable tool for quantitative height estimations, careful consideration of these potential error sources is essential for ensuring the reliability and robustness of the derived height data.

### 4.2. Advantages of High-Resolution CHM Data for Kimchi Cabbage Monitoring

The high-resolution CHM data (0.75 cm) acquired in this study offered several distinct advantages for monitoring Kimchi cabbage growth, consistent with the findings from other crop monitoring studies utilizing high-resolution imagery [37,38]. The detailed spatial information captured by the CHM enabled the visualization and assessment of individual Kimchi cabbage plants, facilitating the identification of plant-level growth characteristics and the detection of vacant areas in the field. This capability is particularly valuable for the early identification of growth irregularities or potential issues, allowing for timely intervention and improved crop management.

Furthermore, the high-resolution CHM data enabled the precise tracking of the growth dynamics over time, providing insights into the plant responses to environmental factors and management practices. This temporal monitoring is essential for understanding growth patterns and optimizing cultivation strategies to maximize the yield potential. Additionally, the ability to visualize individual plant heights and monitor their growth progression can be leveraged for quality control purposes. By identifying plants that deviate from the expected growth trajectory, growers can implement targeted interventions to ensure uniform crop development and enhance overall crop quality.

In conclusion, the high-resolution CHM data derived from drone imagery proved invaluable for enhancing precision agriculture practices in Kimchi cabbage cultivation. The ability to visualize and analyze plant-level growth, monitor temporal dynamics, and implement targeted interventions based on CHM data contributes to improved crop management, optimized resource allocation, and, ultimately, increased productivity and profitability in Kimchi cabbage farming.

### 4.3. Overestimation of Kimchi Cabbage Heights and Model Performance

This study identified a consistent overestimation of Kimchi cabbage heights in the drone-derived measurements (the CHM). This discrepancy is primarily attributed to the inclusion of neighboring Kimchi cabbage leaves within the area of interest (AOI) during the height extraction. As Kimchi cabbages grow and their leaves expand, they often overlap with neighboring plants, leading to an overestimation of the individual plant height when using a fixed AOI. This effect is more pronounced in the later growth stages when plant density and leaf overlap are highest. This issue, also reported in previous studies [28], underscores the need for improved object segmentation algorithms to accurately delineate the individual plants and their leaves, especially in dense canopies. Future research should focus on developing more sophisticated segmentation techniques to enhance the accuracy of the height estimations.

The relatively low $R^2$ values observed in some of the LSTM models (Table 5) can be attributed to several factors. The inherent variability in Kimchi cabbage growth, influenced by genetic and environmental factors, poses a challenge for precise prediction. Additionally, limitations in the data, such as the number of data points used for the training and the potential errors in the height estimation from the drone imagery, could have contributed to the lower $R^2$ values. Furthermore, the model architecture itself might have room for improvement, and exploring alternative LSTM configurations or incorporating additional features could lead to an enhanced performance. These challenges are not unique to this study; similar observations have been made in other crop growth prediction models using machine learning [5,21,36].

### 4.4. Logistic Function, Growth Stage Prediction, and Practical Applicability

Existing research on crop heights using drone imagery has largely focused on estimating current conditions [37,38]. However, this study stands out for its ability to predict cabbage heights early by utilizing height data from different growth stages. Additionally, the accuracy of the height estimates was improved by interpolating missing height data using growth curves [29–32].

This study successfully applied the logistic function to capture Kimchi cabbage growth patterns and to predict the final height using the data from specific growth stages. The logistic model's ability to accurately represent the sigmoidal growth curve aligns with its successful application in previous crop growth studies [10,33]. The selection of the appropriate growth stages for the data collection proved crucial for accurate predictions, emphasizing the importance of capturing key inflection points in the growth curve [34,35].

While the LSTM model shows promise, its real-world applicability necessitates a thorough evaluation of economic feasibility and logistical challenges [36]. The costs associated with drone equipment, image processing software, and computational resources need to be considered. Additionally, logistical challenges such as the requirement for skilled drone operators, data management infrastructure, and potential regulatory constraints need to be addressed. Strategies for overcoming these challenges and promoting wider adoption, such as developing user-friendly software and providing training programs, should be explored.

*4.5. Prediction Accuracy, Future Advancements, and the Potential of the LSTM Model*

This study successfully demonstrated the potential of predicting Kimchi cabbage heights solely based on drone imagery, a methodological advancement compared to previous studies that relied on additional data sources [9,10]. While the results are promising, there is room for further improvement in the prediction accuracy. Limitations in image resolution and Kimchi cabbage object segmentation techniques can introduce errors in the height estimation, which may propagate to the final prediction. As drone technology and image processing algorithms continue to advance, we can expect significant improvements in both image resolution and object segmentation accuracy. Higher resolution images would allow for more precise measurement of the plant height, while improved segmentation algorithms would enable more accurate identification of the individual Kimchi cabbage plants and their leaves, even in dense canopies [37,38].

The LSTM model's potential extends beyond Kimchi cabbage height predictions. Incorporating environmental data such as temperature, humidity, and soil moisture, alongside height measurements, could enable more comprehensive growth analysis and improve the prediction accuracy, as demonstrated in other studies [14,39]. The model's adaptability to other crops with similar growth patterns further expands its potential for broader agricultural applications.

Future research should focus on refining the LSTM model by incorporating additional data sources, exploring different model architectures, and validating its performance across diverse environmental conditions and cultivars. By continuously improving and expanding the capabilities of such models, we can unlock their full potential for transforming agricultural practices and promoting sustainable crop production.

## 5. Conclusions

This study successfully developed and validated a novel method for the early prediction of Kimchi cabbage heights using drone imagery and an LSTM-based prediction model. The integration of high-resolution drone imagery, a logistic growth curve interpolation, and LSTM modeling enabled the accurate estimation and prediction of Kimchi cabbage heights at different growth stages. The model trained on data collected 36 days after planting (DAPs) demonstrated the best balance between accuracy (MAE = 2.77 cm) and prediction timing, making it suitable for informing timely cultivation management decisions. The spatial distribution maps generated from the model's predictions further highlighted its potential for site-specific management, revealing distinct growth patterns between different soil types. The model accurately captured the slower growth in loam soil compared to sandy loam.

This research contributes to the field of precision agriculture by providing a practical and efficient method for the early prediction of Kimchi cabbage heights. The proposed technology empowers farmers with data-driven insights, enabling timely interventions, such as fertilization or chemical treatments, based on predicted growth patterns. This proactive approach can lead to improved crop yields, resource optimization, and, ultimately, increased profitability.

Future research will focus on refining the model's accuracy by developing advanced object segmentation algorithms and incorporating environmental data such as temperature, humidity, and soil moisture. The applicability of the LSTM model to other crops will also be explored, further expanding the potential of precision agriculture technologies.

In conclusion, this study highlights the transformative potential of integrating drone technology, data analytics, and AI in agriculture. By enabling the early and accurate prediction of Kimchi cabbage heights, this research paves the way for a more efficient, sustainable, and profitable future for farmers and the agricultural industry.

## 6. Patents

This paper has nothing to do with applications for intellectual property rights such as patents.

**Author Contributions:** Conceptualization, S.-h.G.; methodology, S.-h.G.; software, S.-h.G.; validation, J.-h.P.; formal analysis, S.-h.G. and J.-h.P.; investigation, S.-h.G. and J.-h.P.; resources, S.-h.G.; data curation, S.-h.G.; writing—original draft preparation, S.-h.G.; writing—review and editing, J.-h.P.; visualization, S.-h.G.; and supervision, J.-h.P. All authors have read and agreed to the published version of this manuscript.

**Funding:** This study received no funding.

**Data Availability Statement:** The original contributions presented in the study are included in the article, further inquiries can be directed to the corresponding author.

**Conflicts of Interest:** The authors declare no conflicts of interest.

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
