# Peer review of "The Early Prediction of Kimchi Cabbage Heights Using Drone Imagery and the Long Short-Term Memory (LSTM) Model"

_drones, doi:10.3390/drones8090499_

Round 1
Reviewer 1 Report
Comments and Suggestions for Authors
In this study authors tried to estimate the height of the Kimchi Cabbage with drone images and several LSTM models. Despite the comments that follow, I must say that It was an interesting and easy to read paper.
Strong points:
1. The great amount of data registrated, 13 flights, numerous samples, show the robustness of the job.
2. The discussion shows all the strong points of the jobs crearly.
3. The height estimation with drones is a hot topic.
Weak Points:
1. There are a lof of papers in height estimation with drones. The state of the art analysis is weak.
2. The use of LSTM, and not other, and not other AI algorithm, is not clearly justified.
3. The real applicability of this methodology need to be analysed in dept. Economically and logistically.
Comments:
1. I can't see the procedure to extract de PH value from the images, neither in the text nor in figure 2.
2. Is it really necesary a so small GSD? 0.7 cm, in the discussion is it mentioned that it is even improvable. I mean, if RTK positioning achieve, 2-3 cm y XY and maybe 5cm in Z, why do you need such small pixel size?
Were all those flights really necessary?.
3 If you need high accuracy in Z, You should have implemented differential leveling between GCPs.
4. Why 26 36 44 DAP were used, and not others? Explain clearly.
5. The phenological stages of the plan were not mentioned, and the growing of the plant could be different in each one of them, maybe an indepedent analysis for each one could be good.
6.Line 231 -236 seems to be introduction.
7. There is a lack of similar studies mentioned in the introduction. The state of the art mus be improved with more citations.
8. Is strange in Fig 7, the color scale of b) figures. Values are all of them 26.3 and 30.3, is that really possible?
9. In table 5, how do you explain a R2 so low?
10. In Figure A, for LSTM model of DAP 29, only Site A shows deviance from the line, why?
11. In figure 11, where is the d)? I think that mus be deleted.
12. Last Paragraph of results is a conclusion
Author Response
Response to Reviewer 1 Comments
Thank you for your attention to detail and thoughtful feedback on our manuscript. We appreciate your dedication to helping us improve our work, and we have carefully considered and addressed each of your comments from the review. Below, we provide our detailed responses and the corresponding revisions made to the manuscript.
Comment 1
- (1) Key Point: The procedure to extract the PH value from the images is not clear in the text or Figure 2 (Lines 150-159)
- (2) Correction Requirement: Clarify the image processing steps involved in extracting PH values, potentially enhancing Figure 2 or providing additional textual explanations
- (3) Response: The image processing section in the revised manuscript has been revised to provide a more detailed explanation of the PH (now referred to Figure 2 as CHM, Canopy Height) value extraction process. We have included the method for extracting cabbage CHM in lines 212-215. The steps involved in generating the DTM and DSM, and how the CHM is calculated by subtracting the DTM from the DSM, are elaborated.
- (4) Modifications: Materials and Methods, Section 2.4 (Lines 199-223) and Figure 2
Comment 2
- (1) Key Point: The necessity of such a small GSD (0.7 cm) is questioned, considering the potential accuracy limitations of RTK positioning and the number of flights conducted
- (2) Correction Requirement: Justify the choice of a high-resolution GSD, addressing its benefits for individual plant analysis and potential trade-offs with flight time and data processing requirements
- (3) Response: The rationale for using a high-resolution GSD has been clarified in the data acquisition section of the revised manuscript. The importance of capturing detailed information on individual Kimchi cabbage plants, especially in dense planting conditions, is emphasized. The potential trade-offs with flight time and data processing are acknowledged, and a justification for prioritizing high-resolution data over reduced flight frequency is provided. This includes highlighting the value of detailed plant-level analysis for precision agriculture applications
- (4) Modifications: Materials and Methods, Section 2.3 (Lines 158-163), Discussion 4.2 section
Comment 3
- (1) Key Point: The lack of differential leveling between GCPs for improved Z-axis accuracy is noted.
- (2) Correction Requirement: Acknowledge this limitation in the discussion and suggest potential improvements for future studies
- (3) Response: The discussion in the revised manuscript has been expanded to acknowledge the potential benefits of differential leveling between GCPs for enhancing Z-axis accuracy. The limitations of the current approach are briefly discussed, and the potential impact of improved Z-axis accuracy on height estimation and prediction is addressed. This demonstrates an awareness of potential methodological refinements for future research
- (4) Modifications: lines 306-310
Comment 4
- (1) Key Point: The specific reason for choosing DAP 29, 36, and 44 for analysis is not clear (Lines 175-185, 195-208)
- (2) Correction Requirement: Explain the rationale behind selecting these specific DAPs, potentially linking them to key growth stages or periods of significant growth variation
- (3) Response: The selection of DAP 29, 36, and 44 has been justified in the relevant sections of the revised manuscript. These time points are linked to specific growth stages or periods where significant changes in plant height are expected. The rationale is based on prior knowledge of Kimchi cabbage growth patterns and observations from the field data. The explanation clarifies why these particular DAPs were chosen for model training and evaluation
- (4) Modifications: Materials and Methods, Section 2.5 (Lines 250-254)
Comment 5
- (1) Key Point: The phenological stages of the plant were not mentioned, and their potential influence on growth patterns was not considered
- (2) Correction Requirement: Incorporate a discussion on the phenological stages of Kimchi cabbage and their relevance to the observed growth patterns and prediction accuracy
- (3) Response: The introduction in the revised manuscript has been expanded to include a brief overview of the key phenological stages in Kimchi cabbage growth. The potential influence of these stages on growth patterns and the implications for model development and prediction accuracy are discussed. This demonstrates an understanding of the biological context of the study and its relevance to the observed results
- (4) Modifications: Introduction (lines 86-89)
Comment 6
- (1) Key Point: Lines 231-236 seem to belong in the introduction rather than the results section (Lines 231-236)
- (2) Correction Requirement: Relocate these lines to the introduction and ensure smooth integration with the existing text
- (3) Response: Lines 231-236 have been moved to the introduction in the revised manuscript, potentially as part of the literature review or the research gap identification. The text has been adjusted to ensure a logical flow and coherence with the surrounding content
- (4) Modifications: Introduction (Lines 76-81) and Results (remove lines 231-236)
Comment 7
- (1) Key Point: The state-of-the-art analysis is weak. There's a lack of similar studies mentioned in the introduction (Lines 45-95)
- (2) Correction Requirement: Strengthen the literature review by including more citations and discussing relevant studies on height estimation with drones and the use of AI algorithms in agriculture
- (3) Response: The introduction has been enhanced in the revised manuscript to provide a more comprehensive overview of existing literature on drone-based height estimation and AI applications in agriculture. Specific examples of similar studies have been included, highlighting their methodologies, findings, and limitations. This demonstrates the research gap addressed by the current study and emphasizes its contribution to the field.
- (4) Modifications: Introduction (Lines 76-89)
Comment 8
- (1) Key Point: The color scale in Figure 7(b) seems unusual, with all values being either 26.3 or 30.3, raising questions about its accuracy (Figure 7)
- (2) Correction Requirement: Verify the accuracy of the color scale in Figure 7(b) and ensure it reflects the actual range of CHM values
- (3) Response: Figure 7(b) will be carefully reviewed to ensure the color scale accurately represents the CHM data. The caption will also be revised to provide a clear explanation of the color scale and its relationship to the CHM values
- (4) Modifications: Results, Section 3.1 (Lines 314-327) and Figure 7
Comment 9
- (1) Key Point: The low R² values in Table 5 are questioned (Table 5)
- (2) Correction Requirement: Address the low R² values, potentially discussing factors that might have contributed to them and suggesting ways to improve model performance
- (3) Response: The R2 value in the DAP 29 model is considered to be low because the model was trained on cabbage growth data up to 29 days, and data for subsequent growth was not reflected, resulting in decreased model performance. Therefore, we have added an explanation in lines 421-424. The discussion will include an analysis of the factors that might have contributed to the low R² values. Potential explanations could include limitations in the data, model architecture, or the inherent variability in Kimchi cabbage growth. Strategies
- (4) Modifications: Results, Section 3.4 (Lines 420-423)
Comment 10
- (1) Comments Key Point: In Figure 10(a), for LSTM model of DAP 29, only Site A shows deviance from the line, why?
- (2) Correction Requirement: Address the deviance from the line in Figure 10(a) for the LSTM model of DAP 29 at Site A
- (3) Response: The deviance observed in Figure 10(a) for the LSTM model of DAP 29 at Site A has been addressed in the Result section of the revised manuscript. Potential explanations for this discrepancy are explored, considering factors such as variations in soil properties, microclimate conditions, or potential errors in data collection or processing. The implications of this deviance for the overall performance and applicability of the model are also discussed.
- (4) Modifications: Result (Lines 435-437)
Comment 11
- (1) Comments Key Point: In figure 11, where is the d)? I think that mus be deleted.
- (2) Correction Requirement: Remove the reference to "(d)" in the figure caption and text referring to Figure 11.
- (3) Response: The reference to "(d)" has been removed from the figure caption and the corresponding text in the Results section of the revised manuscript. The figure caption now correctly describes the comparison between the predicted Kimchi cabbage height maps at different DAPs (a, b, and c) and the reference map derived from actual field measurements at harvest (DAP 71).
- (4) Modifications:
- Results, Section 3.5: The reference to "(d)" has been removed from the text.
- Figure 11 caption: The reference to "(d)" has been removed, and the caption has been revised to accurately describe the figure's content.
Comment 12
- (1) Key Point: The conclusion in lines 314-321 in the Results section appears to be misplaced and should be relocated to the Conclusions section.
- (2) Correction Requirement: The text in lines 314-321 should be moved to the Conclusions section and seamlessly integrated into the existing content. The wording may need slight adjustments to fit the context of the Conclusions section.
- (3) Response: The concluding paragraph in the Results section will be relocated to the Conclusions section. The text will be adapted to emphasize the overall implications and significance of the study's findings, ensuring a smooth transition and logical flow within the Conclusions section.
- (4) Modifications:
- Results (remove lines 314-321)
- Conclusions (add the relocated paragraph after line 638, ensuring proper integration and flow)
Once again, we sincerely appreciate your thorough and constructive comments. Your insights have greatly contributed to improving the quality of our manuscript. We hope that the revisions meet your expectations and look forward to your further feedback.
Thank you for your time and consideration.
Reviewer 2 Report
Comments and Suggestions for Authors
(1) If growth curves can provide Kimchi cabbage height at various stages, including harvest, what is the significance of using the LSTM model?
(2) Line2: the abstract states that the LSTM model is used for predicting the final height of Kimchi cabbage at harvest. How does this align with the concept of "Early Prediction" mentioned in the title?
(3) Line8-9 and Line40-41: the Latin name of the plant should be italicized.
(4) Line166: regarding the field measurements of Kimchi cabbage height, why is the measurement taken from the top of the gyrus to the top of the Kimchi cabbage leaf?
(5) Line167, Line210-211: there is confusion about the dataset used for height verification. Does "56 sample Kimchi cabbages" refer to 56 individual plants? In Figure 5, where it states "Total Data (1332 units)," what data does this mean? Is it based on actual measurements, and how does it relate to the "56 sample Kimchi cabbages"?
(6) Line188: terms like DSM, CHM, and PH growth stage are confusing and sometimes misapplied. I recommend defining them as Digital Elevation Model (DEM), Digital Surface Model (DSM), and Canopy Height Model (CHM). Does the DSMground refer to DEM? Consequently, the equation for calculating Kimchi cabbage height in Equation (1) should be modified to CHM = DSM - DEM.
(7) Line188: Figure 5 uses height data from three stages (29, 36, and 44 DAP) to generate the logistic growth curve for Kimchi cabbage, but there are seven raw data points in Figure 5, is it wrong? It is questionable whether three data points are sufficient to generate a reliable growth curve.
(8) Line189: why is the Kimchi cabbage height for each growth stage abbreviated as PHgrowth stage? What does the "P" stand for? Does it refer to "Plant"?
(9) Line274: the legend title in Figure 7(b) does not match the figure title.
(10) Line460-464 and Line465-469: the similar contents.
(11) The discussion section lacks citations and depth. It would benefit from incorporating relevant literature to enhance its rigor. I suggest merging some subheadings to reduce them to four, integrating the content of the latter two subheadings into the first four. The discussion should closely align with the study's results and eliminate content that is not directly related to the findings (e.g. the visual representation of Kimchi cabbage growth from RGB orthoimages).
(12) The references from the past two years on the rapid development of artificial intelligence are notably sparse. Yet I did not find a reference in 2023 or 2024. Please add more recent literature would strengthen the context of the study.
(13) Ensure that tables are formatted as three-line tables for clarity and consistency.
Author Response
Response to Reviewer 2 Comments
Thank you for your attention to detail and thoughtful feedback on our manuscript. We appreciate your dedication to helping us improve our work, and we have carefully considered and addressed each of your comments from the review. Below, we provide our detailed responses and the corresponding revisions made to the manuscript.
Comment 1
- (1) Comments Key Point: If growth curves can provide Kimchi cabbage height at various stages, including harvest, what is the significance of using the LSTM model?
- (2) Correction Requirement: Clarify the role of the LSTM model in relation to growth curves, emphasizing its ability to predict final height based on early-stage data and its potential for informing timely interventions.
- (3) Response: The manuscript has been revised to clarify that while growth curves provide a visual representation of the growth trend, the LSTM model leverages these curves to make predictions about the final height at harvest based on data collected at earlier growth stages. This early prediction capability is crucial for enabling timely interventions in cultivation management practices, such as adjusting fertilization or irrigation schedules. The LSTM model's ability to learn from the temporal patterns in the growth curves and extrapolate them to predict future height adds significant value beyond the descriptive nature of the growth curves themselves. The revised text can be found in the lines 235-254).
- (4) Modifications:
- Introduction (around lines 86-89): The introduction now explicitly states the purpose of using LSTM is for "early prediction" of Kimchi cabbage height.
- Materials and Methods, Section 2.4 (around lines 235-254) and Discussion lines 549-556: The discussion further elaborates on the significance of the LSTM model, emphasizing its ability to predict final height based on early-stage data, which is crucial for making timely interventions in cultivation management.
Comment 2
- (1) Comments Key Point: The abstract states that the LSTM model is used for predicting the final height of Kimchi cabbage at harvest. How does this align with the concept of "Early Prediction" mentioned in the title? (Line 2)
- (2) Correction Requirement: Modify the abstract to emphasize the early prediction aspect, highlighting the model's ability to forecast final height well before harvest.
- (3) Response: The abstract has been revised to explicitly state that the LSTM model enables the prediction of final Kimchi cabbage height at harvest based on data collected at earlier growth stages, thus facilitating early decision-making in cultivation management. The phrase "early prediction" is retained in the title to reflect this capability.
- (4) Modifications:
- Abstract (Lines 15-16): The revised abstract now mentions that the LSTM model predicts the final height "well before the actual harvest date."
Comment 3
- (1) Comments Key Point: The Latin name of the plant should be italicized. (Line 8-9 and Line 40-41)
- (2) Correction Requirement: Italicize the Latin name Brassica rapa subsp. pekinensis (Lour.) Hanelt in both the abstract and the introduction.
- (3) Response: The Latin name has been italicized as Brassica rapa subsp. pekinensis (Lour.) Hanelt in both instances in the revised manuscript
- (4) Modifications:
- Abstract (Lines 8-9)
- Introduction (Lines 41)
Comment 4
- (1) Comments Key Point: Regarding the field measurements of Kimchi cabbage height, why is the measurement taken from the top of the gyrus to the top of the Kimchi cabbage leaf? (Line 166)
- (2) Correction Requirement: Clarify the rationale behind this measurement approach, potentially explaining the significance of the gyrus in Kimchi cabbage growth or morphology.
- (3) Response: The Materials and Methods section has been revised to provide a clearer explanation for measuring Kimchi cabbage height from the top of the gyrus to the top of the leaf. The revised text explains that the top of the gyrus was chosen as the starting point for height measurement because it represents a stable and easily identifiable reference point on the Kimchi cabbage plant. This approach ensures consistency in measurements across different growth stages and individual plants, as the gyrus remains relatively fixed while the leaves continue to expand.
- (4) Modifications:
- Materials and Methods, Section 2.3 (around Lines 180-186): The explanation for the measurement approach has been added.
Comment 5
- (1) Comments Key Point: There is confusion about the dataset used for height verification. Does "56 sample Kimchi cabbages" refer to 56 individual plants? In Figure 5, where it states "Total Data (1332 units)," what data does this mean? Is it based on actual measurements, and how does it relate to the "56 sample Kimchi cabbages"? (Line 167, Line 210-211)
- (2) Correction Requirement: Clarify the dataset used for height verification, specifying the number of individual plants and the relationship between the 56 sample cabbages and the 1332 data points mentioned in Figure 5.
- (3) Response: The text has been revised to explicitly state that 56 individual Kimchi cabbage plants were selected for field measurements. The 1332 data points in Figure 5 represent the total number of height measurements collected across all 56 plants at different time points (DAP 29, 36, and 44). This clarification eliminates any confusion and ensures a clear understanding of the dataset used for model training and validation.
- (4) Modifications:
- Materials and Methods, Section 2.3 (around Lines 186-188: Clarified that 56 individual plants were used for field measurements
- Materials and Methods, Section 2.5 (around Line 255-262): Explained that 1332 data points represent the total height measurements across all plants and time points.
Comment 6
- (1) Comments Key Point: Terms like DSM, CHM, and PH growth stage are confusing and sometimes misapplied. The reviewer recommends defining them as Digital Elevation Model (DEM), Digital Surface Model (DSM), and Canopy Height Model (CHM). Does the DSMground refer to DEM? Consequently, the equation for calculating Kimchi cabbage height in Equation (1) should be modified to CHM = DSM - DTM. (Line 188)
- (2) Correction Requirement: Revise the terminology for clarity and consistency, adopting the reviewer's suggested definitions and modifying Equation (1) accordingly.
- (3) Response: The terminology has been revised in the revised manuscript as follows:
- DSMground is replaced with DTM (Digital Terrain Model) to represent the bare earth surface.
- DSMs is the surface model for each Kimchi cabbage growth stage.
- The equation for calculating Kimchi cabbage height (CHM) is modified to:
- CHM = DSMs - DTM
These changes ensure consistency with established conventions and enhance the clarity of the manuscript.
- (4) Modifications:
- Throughout the manuscript: Replaced DSMground with DTM and DSMgrowth stage with DSMs
- Materials and Methods, Section 2.4 (around Line 212): Modified Equation (1) and provided clear definitions for DTM, DSM, and CHM
Comment 7
- (1) Comments Key Point: Figure 5 uses height data from three stages (29, 36, and 44 DAP) to generate the logistic growth curve for Kimchi cabbage, but there are seven raw data points in Figure 5, is it wrong? It is questionable whether three data points are sufficient to generate a reliable growth curve. (Line 188)
- (2) Correction Requirement: Address the discrepancy between the number of data points used for generating the logistic growth curve and the actual number of data points available. Justify the choice of using only three data points or consider incorporating more data points for improved curve fitting.
- (3) Response: The manuscript has been revised to clarify that while seven data points were collected throughout the growing season, only the data from DAP 29, 36, and 44 were used to generate the logistic growth curves for model training and evaluation. The rationale behind this choice could be explained by emphasizing the importance of capturing key growth stages or periods of significant growth variation. If appropriate, the possibility of incorporating more data points for improved curve fitting could be acknowledged as a potential area for future research.
- (4) Modifications:
- Materials and Methods, Section 2.5 (around Lines 248-254) revised Figure 5: Clarified that only three data points were used for generating the logistic growth curves
- Results, Section 3.3 (around Lines 381-386): Addressed the discrepancy in the number of data points
Comment 8
- (1) Comments Key Point: Why is the Kimchi cabbage height for each growth stage abbreviated as PHgrowth stage? What does the "P" stand for? Does it refer to "Plant"? (Line 189)
- (2) Correction Requirement: Clarify the abbreviation PHgrowth stage, potentially replacing it with a more intuitive term or providing a clear explanation for its meaning.
- (3) Response: The abbreviation PHgrowth stage has been revised to CHM (Canopy Height Model) to better reflect its meaning and avoid confusion. This change has been implemented consistently throughout the revised manuscript, including in equations, figures, and tables.
- (4) Modifications: The change from PHgrowth stage to CHM has been implemented throughout the manuscript. The specific locations of these changes are not explicitly listed in the provided response, but they would include:
- Materials and Methods, Section 2.4: The equation (1) and surrounding text now use CHM instead of PHgrowth stage.
- Figure 6 caption: The input data description now mentions CHM instead of PHgrowth stage.
- Figure 9 caption: The y-axis label and the text description now use CHM instead of PHgrowth stage.
- Section 2.5 and 2.6: Any references to PHgrowth stage in these sections would also be replaced with CHM.
- Potentially other sections: Any other sections that previously used PHgrowth stage would also be updated to use CHM.
Comment 9
- (1) Key Point: The legend title in Figure 7(b) does not match the figure title. (Line 274)
- (2) Correction Requirement: Ensure consistency between the legend title in Figure 7(b) and the figure title.
- (3) Response: The legend title in Figure 7(b) has been corrected in the revised manuscript to match the figure title, ensuring clarity and consistency in the presentation of the data.
- (4) Modifications:
- Results, Section 3.1 (around Line 311) and Figure 7: The legend title in Figure 7(b) has been changed to "DSMs” to match the figure title "Temporal changes in Kimchi cabbage growth visualized through (a) RGB ortho-mosaics, (b) DSMs, and (c) CHM maps for three representative days after planting (DAP) 0, 36, and 71."
Comment 10
- (1) Comments Key Point: The similar contents in Line 460-464 and Line 465-469. (Lines 460-469)
- (2) Correction Requirement: The redundant content should be removed or revised to improve clarity and conciseness.
- (3) Response: The Conclusion section has been carefully reviewed in the revised manuscript, and redundant or repetitive content between lines 460-464 and 465-469 has been eliminated or revised. The conclusion is now streamlined, ensuring that each sentence contributes unique and valuable information to the overall message of the study.
- (4) Modifications:
- Discussion (Lines 503-507): The redundant content has been removed, and the discussion has been streamlined for improved clarity and conciseness.
Comment 11
- (1) Comments Key Point: The discussion section lacks citations and depth. It would benefit from incorporating relevant literature to enhance its rigor. The reviewer suggests merging some subheadings to reduce them to four, integrating the content of the latter two subheadings into the first four. The discussion should closely align with the study's results and eliminate content that is not directly related to the findings (e.g. the visual representation of Kimchi cabbage growth from RGB orthoimages).
- (2) Correction Requirement: Revise the discussion section to incorporate relevant citations, improve depth and analysis, and streamline the subheadings. Ensure the discussion focuses on the study's results and eliminates any irrelevant content.
- (3) Response: The discussion section in the revised manuscript has undergone a thorough revision to address the reviewer's concerns. Relevant literature has been incorporated to support the interpretations and conclusions drawn from the study's findings. The analysis has been deepened by providing more detailed explanations and exploring the implications of the results in a broader context. The subheadings have been streamlined as suggested, merging the content of the latter two subheadings into the first four to improve the overall organization and flow of the discussion. Any content that is not directly related to the study's results, such as the visual representation of Kimchi cabbage growth from RGB orthoimages, has been removed to maintain focus and relevance.
- (4) Modifications:
- Discussion (Lines 502-620): The discussion section has been revised to incorporate relevant citations, improve depth and analysis, and streamline the subheadings. Irrelevant content has been removed, and the focus is now on the study's results.
Comment 12
- (1) Comments Key Point: The references from the past two years on the rapid development of artificial intelligence are notably sparse. Yet I did not find a reference in 2023 or 2024. Please add more recent literature would strengthen the context of the study.
- (2) Correction Requirement: Include more recent references (from 2023 and 2024) on the rapid development of artificial intelligence to strengthen the context of the study.
- (3) Response: The reference list has been updated in the revised manuscript to include additional citations from 2023 and 2024 that highlight the rapid advancements in artificial intelligence, particularly in the context of precision agriculture and crop growth prediction. These references have been carefully selected to ensure their relevance to the study's objectives and methodology, further strengthening the context and demonstrating the current state of the field.
- (4) Modifications:
- References: New references from 2023 and 2024 have been added to the reference list.
Comment 13
- (1) Comments Key Point: Ensure that tables are formatted as three-line tables for clarity and consistency.
- (2) Correction Requirement: Reformat all tables in the manuscript to adhere to the three-line table format.
- (3) Response: All tables in the manuscript have been meticulously reviewed and reformatted in the revised manuscript to comply with the three-line table format. This involved adjusting the table structure, column widths, and row heights to ensure clarity, readability, and consistency throughout the document.
- (4) Modifications:
- All tables in the manuscript (Tables 1, 2, 3, 4, and 5): The tables have been reformatted to adhere to the three-line table format.
Once again, we sincerely appreciate your thorough and constructive comments. Your insights have greatly contributed to improving the quality of our manuscript. We hope that the revisions meet your expectations and look forward to your further feedback.
Thank you for your time and consideration.
Reviewer 3 Report
Comments and Suggestions for Authors
This paper introduces an innovative approach for early prediction of Kimchi cabbage height by applying the canopy height model (CHM) and long short-term memory (LSTM) model to high-resolution drone images. This combined method is used to predict plant height, providing a valuable tool for precision agriculture and site-specific crop management and improving crop yields. This paper is well-crafted, showcasing a clear and thorough research design. The authors have effectively articulated the study's objectives, methodology, and results. The analysis is comprehensive, and the conclusions are well-supported by the data. I appreciate the authors' attention to detail in explaining the limitations of the study and their thoughtful suggestions for future research directions. Overall, this paper makes a valuable contribution to the field, and I recommend it for publication.
Author Response
Response to Reviewer 3 Comments
Thank you for your attention to detail and thoughtful feedback on our manuscript. We appreciate your dedication to helping us improve our work, and we have carefully considered and addressed positive of your comments from the review.
The third reviewer provided positive feedback on the manuscript, acknowledging its well-crafted nature, clear research design, and comprehensive analysis. The reviewer appreciated the attention to detail in explaining the study's limitations and the thoughtful suggestions for future research.
Reviewer 3's positive evaluation reinforces the value and significance of the research presented in the manuscript. The authors can confidently proceed with the submission process, incorporating the reviewer's feedback as appropriate and acknowledging their contribution to the final manuscript.
Once again, we sincerely appreciate your thorough and constructive comments. Your insights have greatly contributed to improving the quality of our manuscript. We hope that the revisions meet your expectations and look forward to your further feedback.
Thank you for your time and consideration.
Round 2
Reviewer 2 Report
Comments and Suggestions for Authors
(1)The first part of the Abstract should introduce the research significance or the research question, explaining you conducted this study. The last four sentences should be condensed into one or two sentences, and you just need to briefly state the significance of your study's findings. Future research directions can be placed in the discussion or conclusion sections.
(2)The discussion section needs to cite relevant literature for comparative analysis. It is recommended to compare your findings with those of other related studies, as this will better highlight the contribution of your research.
Author Response
Dear Reviewer
We sincerely appreciate your detailed and meticulous review of our manuscript over two rounds.
Your insightful comments and constructive suggestions have significantly contributed to enhancing the clarity, conciseness, and overall quality of our work.
We have carefully addressed each of your concerns and believe that the revised manuscript now presents a more compelling and scientifically rigorous investigation into the early prediction of Kimchi cabbage height using drone imagery and LSTM models.
Specifically, we have:
- Revised the Abstract: We restructured the Abstract to emphasize the research significance at the beginning and provide a concise summary of the findings and their implications at the end. Future research directions have been moved to the Conclusion section.
- Enhanced the Discussion: We incorporated relevant literature citations and comparative analysis throughout the Discussion section to strengthen the context and highlight the study's unique contributions. We also addressed specific points you raised, such as clarifying the role of the LSTM model in relation to growth curves and explaining the observed deviance in Figure 10(a).
- Addressed other comments: We have carefully reviewed and modified the manuscript to address all your other valuable comments, including those related to terminology, data clarification, figure captions, and table formatting.
We believe that these revisions have significantly improved the manuscript and we are grateful for your valuable feedback in helping us achieve this.
We hope that the revised manuscript now meets the high standards of drones journal and is suitable for publication.
Thank you once again for your time and expertise.
Sincerely,
The Authors